

# Applying foliar stoichiometric traits of plants to determine fertilization for a mixed pine-oak stand in the Qinling Mountains, China

Lin Hou[1], Zhenjie Dong[1], Yuanyuan Yang[1], Donghong Zhang[1], Shengli Zhang[2] and Shuoxin Zhang[1]

[1] College of Forestry, Northwest A&F University, Yangling, Shaanxi, China
[2] College of Natural Resources and Environment, Northwest A&F University, Yangling, Shaanxi, China

## ABSTRACT

**Background.** The Chinese Natural Forest Protection program has been conducted nationwide and has achieved resounding success. However, timber importation has increased; therefore, producing more domestic timber is critical to meet the demand for raw materials. Fertilization is one of the most effective silviculture practices used to improve tree and stand growth. However, determining the appropriate type and amount of elements is necessary for effective fertilization of big timber in different forest types and environmental conditions. Stoichiometric theory provides the criteria to assess nutrient limitation in plants and offers important insight into fertilizer requirements of forested ecosystems.

**Methods.** Nitrogen (N) and phosphorus (P) concentrations in plants' leaves, mineral soil, and litter were investigated in a mixed pine-oak stand.

**Results.** The big timber rate for *Pinus tabuliformis*, *Pinus armandii* and *Quercus aliena* var. *acutesserata* is 57.71%, 22.79% and 2.78% of current existing individuals respectively. Foliar N and P concentrations were 9.08 and 0.88 mg g$^{-1}$, respectively. The N:P in the plants was 10.30. N concentration and N:P in mineral soil decreased from 0–30 cm soil depth. For litter, N and P concentrations were 16.89 and 1.51 mg g$^{-1}$, respectively, and N:P was 11.51. Concentrations of N and P in mineral soil and litter did not significantly affect plants' leaf concentrations. Similar result was also obtained between litter and mineral soil concentrations. Nitrogen storage in mineral soil was significantly correlated with foliar N:P in the plants.

**Discussion.** Foliar N:P of dominant tree species and the plants, and foliar N concentration in *Pinus tabuliformis* and *P. armandii*, and foliar P concentration of *P. armandii* in the mixed pine-oak stand was lower than that in Chinese and other terrestrial plants. Foliar nutrients in the plants were not affected by soil nutrients. According to the criteria of nutrient limitation for plants, growth of dominant tree species was N limited; therefore, 1.49 t ha$^{-1}$ pure N should be added to forest land to as fertilizer.

Corresponding author
Lin Hou,
houlin_1969@nwsuaf.edu.cn,
houlin1969@163.com

## INTRODUCTION

The Natural Forest Protection Program (NFPP) was implemented in China in 1998 and has achieved or surpassed its initial goal by prohibiting commercial logging and partial or full harvesting of timber (*Xu, 2011*). However, the effectiveness of the NFPP has been disputed as it has increased timber imports and caused deterioration in the structures of forest stands (*Wang et al., 2013*). To redress the insufficient production of domestic timber, the project "Techniques for big timber (diameter at breast height, DBH> 26 cm) cultivation" has been conducted in both northern and southern forests in China. Besides proper forest management (*Hou et al., 2017*), applying fertilizer is also an efficient way to achieve "big timber". However, determining the appropriate type and amounts of elements is necessary in the cultivation of big timbers in different forest types and environmental conditions.

Carbon (C), nitrogen (N), and phosphorus (P) are essential elements for plant growth and metabolic processes (*Yang et al., 2015*). The relative concentrations of C, N, and P in plants is known as stoichiometry (*He et al., 2006*), which is a unifying conceptual framework to examine how proportions of elements affect organisms and ecosystems (*Danger, Gessner & Bärlocher, 2016*). The most common limiting elements, C, N, and P, either individually or in combination, are widespread in terrestrial ecosystems (*Venterink, Van der Vliet & Wassen, 2001*; *Venterink et al., 2003*; *Vitousek et al., 2010*). Stoichiometric traits, particularly C:N and N:P, are useful indicators of nutrient limitation in both terrestrial ecosystems and ecosystem functioning (*Elser et al., 2007*; *Gundersen et al., 2009*).

In the last two decades, most research in ecological stoichiometry has focused on the causes and consequences of variation in C, N, and P ratios in organisms and their resources; however, there are large disparities in knowledge among taxa, ecosystem types, and specific research topics (*Bell et al., 2014*; *Fanin et al., 2013*; *Han et al., 2005*; *He et al., 2006*; *Knoll et al., 2009*; *Sardans, Rivas-Ubach & Peñuelas, 2011*; *Scharler et al., 2015*; *Van Huysen, Perakis & Harmon, 2016*; *Venterink & Güsewell, 2010*; *Vrede et al., 2004*; *Zhan et al., 2017*). Leaf stoichiometry of plants, especially N and P, is very important in analyzing the composition, structure, and function of a community and ecosystem (*Gao, Yu & He, 2013*; *Han et al., 2005*; *Rong et al., 2015*; *Venterink & Güsewell, 2010*). Determining how nutrients limit plant growth and N:P in leaves has become a hot topic (*He et al., 2008*). Previous studies have documented N:P in plant leaves and biomass to infer or assess the degree of N or P limitation at the community level (*Aerts & Chapin III, 1999*; *Ellison, 2006*; *Güsewell, 2004*; *Koerselman & Meuleman, 1996*; *Sardans, Rivasubach & Peñuelas, 2012*). However, determining the fertilizer of a specific community via ecological stoichiometry theory by assessing N and P limitation is still in its infancy (*Finn et al., 2016*; *Hong, Wang & Wu, 2015*). We hypothesized that growth of a dominant tree species would not be limited by nutrients when foliar N:P in the plants meet a threshold (*Aerts & Chapin III, 1999*; *Güsewell, 2004*; *Koerselman & Meuleman, 1996*; *Sardans, Rivasubach & Peñuelas, 2012*). We asked whether the amount of fertilization needed for a concrete forest stand could be determined using current N and P storage in mineral soil.

The Qinling Mountains are located in central China and have historically played a vital role in supplying timber for construction. The mixed pine-oak stand is extensive at

the mid-altitudinal gradient of the Qinling Mountains (*Hou et al., 2017*). Maximizing the volume of this forest type not only increases its carbon sequestration and ecosystem services, but also provides more timber for harvesting. The objectives of this study were to address key knowledge gaps, including: (1) foliar N and P stoichiometric traits in the plants in mixed pine-oak stand; (2) the correlation between N and P stoichiometry in forest soil and leaves in the plants; (3) nutrient limitation of the plants; and (4) preliminary recommendations for a fertilizer for dominant trees growing normally based on stoichiometric traits.

## MATERIALS AND METHODS

### Site description

Experiments were conducted at the Qinling National Forest Ecosystem Research Station (QNFERS), located on the southern slope of the Qinling Mountains, Huoditang, Ningshan County, Shaanxi Province, China (32°18′N, 108°20′E). The altitude of the study area was from 1500 to 2500 m. The area has subtropical climate, with annual mean air temperatures around 8–10 °C, annual mean precipitation around 900–1,200 mm, and annual mean evaporation around 800–950 mm. The main soil type was mountain brown soil, developed from granite material, ranging from 30–50 cm depth. The total forest area in the station was 2,037 hectares. Natural forest occupied 93% of the total forest area in QNFERS, with various vegetation types distributed along an altitudinal gradient, such as evergreen deciduous mixed forest (mixed pine-oak forest), deciduous broad-leaved forest (oak, red birch), temperate coniferous forest (Chinese red pine, Armand pine), and cold temperate coniferous forest (spruce, fir). The most dominant forest type was mixed pine-oak forest restored after rotation felling between 1950 and 1998 with an average stand age of 42 years and an average height of 9.2 m. Common tree species included *Pinus tabuliformis*, *P. armandii*, and *Quercus aliena* var. *acuteserrata*, associated with *Toxicodendron vernicifluum*, *Carpinus turczaninowii*, *Swida macrophylla*, *Acer mono*, *Tilia paucicostata*, *Carpinus cordata*, *Juglans cathayensis*, *Bothrocaryum controversum*, *Rhus potaninii*, *Dendrobenthamia japonica* var. *Chinensis*, *Pterocarya stenoptera*, *Acer davidii*, *Betula albo-sinensis*, *Betula albo-sinensis* var. *septentrionalis*, *Juglans mandshurica*, *Rhus punjabensis* Stewart var. *sinica*, *Prunus padus*, and *Sorbus folgneri*, and understory species were abundant (*Hou et al., 2017*).

Field investigations and sampling were conducted from 10 to 15 September, 2014. Samples of litter and soil were collected from 13 long-term fixed plots in the mixed pine-oak stand. Each plot was 20 × 20 m. Tree species, number, Height (H), and diameter at breast height (DBH) were recorded and used to determine the dominant tree species and DBH classes. Each increase in DBH by 4 cm is a DBH class and mid-diameter is used to stand for it (DBH ranging from 6.1 to 10 cm, the corresponding mid-diameter is 8 cm). One hundred and fifty-six of the average standard trees (AST) in all DBH classes were determined in each of the 13 plots (File S1). Mature, sunlit leaves without disease or insect pests were sampled. Four leaves/needles from each AST were collected in each of the four directions (north, east, south and west) and at different stem heights (crown, intermediate section, and underpart), and mixed into one sample for each AST. Five subplots

$(1 \times 1$ m) within each plot were randomly established to collect litter and mineral soil (depth 0–30 cm, 10 cm each layer). Samples of litter were collected manually and soil samples were collected by an auger (internal diameter 38 mm). Soil volumetric rings $(100 \text{ cm}^3)$ were also used to collect soil samples to measure soil bulk density using the cutting ring method. Five samples of litter and mineral soil from the same depth within each subplot were mixed into one sample. All samples (litter and mineral soil) were weighed in the field before being transported to the laboratory.

## Chemical analysis

Samples of leaves/needles and litter were oven-dried at 60 °C to a constant weight, and then ground using a plant sample mill and sieved through a 1-mm mesh screen. Nitrogen concentrations in leaves/needles and litter were measured using a flow injection analyzer (FIA5000; FOSS, Hoganas, Sweden), while P concentrations were measured using the molybdenum blue colorimetric method after digestion in a $H_2SO_4+H_2O_2$ solution (*Bao, 2000*). Auger soil samples were air-dried under shade, then ground and sieved though a 0.149-mm mesh. Total soil N (TN) was measured using the Kjeldahl method (Kjeltec TM 8400; FOSS, Hoganas, Sweden), and total soil P (TP) was determined using acid digestion in a $H_2SO_4+HClO_4$ solution (*Bao, 2000*).

## Data processing

Importance values (IV) of tree species were calculated following *Busby, Vitousek & Dirzo (2010)*:

$$RH = \frac{\sum H_i}{H} \times 100\% \qquad (1)$$

where, RH is the relative height, $H_i$, is the height of tree species $i$, and $H$ is the height of all tree species in all plots.

$$RF = \frac{\sum F_i}{F} \times 100\% \qquad (2)$$

where, RF is the relative frequency, $F_i$ is the frequency of tree species $i$, and $F$ is the frequency of all tree species in all plots.

$$RD = \frac{\frac{1}{4}\pi f \sum D_i^2}{D^2} \qquad (3)$$

where, RD is the relative basal area, $D_i$ is the DBH of tree species $i$, $D$ is the basal area of all tree species in all plots, and $f$ is the form factor of tree species ($f_{\text{conifer}} = 0.40$, $f_{\text{broadleaf}} = 0.42$).

$$IV = \frac{RH + RF + RD}{3} \times 100\%. \qquad (4)$$

Tree species with IV>10% were determined as dominant.

The big timber rate of dominant tree species was calculated as following.

$$R_{bi} = \frac{N_{bi}}{N_i} \times 100\%$$

where $R_{bi}$ is big timber rate (%) of tree species $i$, $N_{bi}$ is stems of big timber of tree species $i$ and $N_i$ is total stems of tree species $i$ in the plots.

Nitrogen and P stoichiometry (N and P concentration and N:P) of plants' leaves was calculated following *Du et al. (2011)*:

$$S_t = \sum_{i=1}^{n} C_{ni} \times IV_i \tag{5}$$

where, $S_t$ is N and P stoichiometry of all tree species, $C_{ni}$ is the concentration of N or P (mg g$^{-1}$) or N:P in leaves of tree $i$, $IV_i$ is the importance value of tree $i$, and n is the number of dominant tree species.

Nutrient storage (t ha$^{-1}$) in litter was calculated as follows:

$$S_l = C_l B_l \tag{6}$$

where, $S_l$ is the litter nutrient storage (t ha$^{-1}$), $C_l$ is element (N or P) concentration (mg g$^{-1}$), and $B_l$ is the biomass of litter (t ha$^{-1}$).

Storage of nutrients (N and P) in soil was calculated as follows:

$$S_T = \sum_{i=1}^{n} E_i B_i D_i \tag{7}$$

where, $S_T$ is the soil nutrient storage (t ha$^{-1}$), $E_i$ is the concentration of element $i$ (mg g$^{-1}$), $B_i$ is the bulk density (g cm$^{-3}$) in layer $i$, and $D_i$ is depth $i$ (cm).

Differences of N, P concentrations and N:P between dominant tree species were analyzed using Statistical Analysis of the $T$-Test. The extremely significant difference ($p < 0.01$) , significant difference ($0.01 < p < 0.05$) and not significant difference ($p > 0.05$) between paired variables were demonstrated as different capital letters, different lowercase letters and the same lowercase letter respectively. The correlations between foliar N and P and N:P in the plants, foliar N and P, N:P in the plants and mineral soil, and litter were assessed using Spearman's rank correlation. Significance levels were set at $p = 0.05$. All statistical analyses were performed using SPSS software (version 19.0 for Windows; SPSS Inc., Chicago, IL, USA). Figures were plotted using Origin 8.0 (OriginLab Corporation, Northampton, MA, USA).

## RESULTS

### Dominant tree species

There were 21 tree species in the mixed pine-oak stand. Aside from the photophilous species (*P. tabuliformis*, *P. armandii*, and *Q. aliena* var.*acuteserrata*), most of the species were shade-tolerant (*Toxicodendron vernicifluum*, *Carpinus turczaninowii*) or neutral with respective to light (*Acer mono*, *Sorbus folgneri*) (Table 1). The dominant tree species with importance values >10% were *P. tabuliformis*, *P. armandii*, and *Q. aliena* var.*acuteserrata* (Table 1).

The big timber rate for *P. tabuliformis*, *P. armandii*, and *Q. aliena* var. *acuteserrata* was 57.71%, 22.79% and 2.78% of current existing individuals respectively (Table 2).

**Table 1 Importance value of tree species.**

| Species | RH (%) | RD (%) | RF (%) | IV (%) |
|---|---|---|---|---|
| *Pinus tabuliformis* | 30.9 | 65.2 | 9.4 | 35.2 |
| *Pinus armandi* | 24.1 | 27.5 | 8.7 | 20.1 |
| *Quercus aliena* var. *acuteserrata* | 16.3 | 5.1 | 10.2 | 10.6 |
| *Toxicodendron vernicifluum* | 7.0 | 1.3 | 8.7 | 5.6 |
| *Carpinus turczaninowii* | 2.2 | 0.1 | 8.7 | 3.6 |
| *Swida macrophylla* | 5.0 | 0.4 | 5.5 | 3.6 |
| *Acer mono* | 2.0 | 0.1 | 7.9 | 3.3 |
| *Tilia paucicostata* | 1.6 | 0.1 | 6.3 | 2.6 |
| *Carpinus cordata* | 1.3 | 0.0 | 5.5 | 2.3 |
| *Juglans cathayensis* | 1.3 | 0.1 | 4.7 | 2.0 |
| *Bothrocaryum controversum* | 2.2 | 0.1 | 3.1 | 1.8 |
| *Rhus potaninii* | 0.7 | 0.0 | 4.7 | 1.8 |
| *Dendrobenthamia japonica* var. *chinensis* | 0.7 | 0.0 | 3.9 | 1.6 |
| *Pterocarya stenoptera* | 2.2 | 0.1 | 2.4 | 1.5 |
| *Acer davidii* | 1.1 | 0.0 | 3.1 | 1.4 |
| *Betula albo-sinensis* | 0.9 | 0.0 | 3.1 | 1.3 |
| *Betula albo-sinensis* var. septentrionalis | 0.2 | 0.0 | 0.8 | 0.3 |
| *Juglans mandshurica* | 0.2 | 0.0 | 0.8 | 0.3 |
| *Rhus punjabensis Stewart* var. *sinica* | 0.1 | 0.0 | 0.8 | 0.3 |
| *Prunus padus* | 0.1 | 0.0 | 0.8 | 0.3 |
| *Sorbus folgneri* | 0.1 | 0.0 | 0.8 | 0.3 |
| Sum | 100.0 | 100.0 | 100.0 | 100.0 |

**Notes.**
RH, RD, RF and IV in the table represent relative height, relative basal area, relative frequency and important value of tree species, respectively.

## Patterns of foliar N and P and N:P in the plants

Foliar N ($11.84 \pm 2.36$ mg g$^{-1}$ to $21.72 \pm 3.19$ mg g$^{-1}$) and P ($1.20 \pm 0.14$ mg g$^{-1}$ to $2.00 \pm 0.31$ mg g$^{-1}$) and N:P (9.62 to 10.87) exhibited large variation among tree species (Table 3). The general trend demonstrated that foliar nutrients and N:P of *P. tabuliformis* and *P. armandii* were less than that in *Q. aliena* var. *acuteserrata* (Table 3). Moreover, foliar stoichiometric variables of the plants were also less than those in individual tree species in the mixed pine-oak stand (Table 3).

In the plants, concentrations of foliar N and P were significantly, positively correlated ($p = 0.000$) with each other (Fig. 1A). The mathematical Log foliar N concentration was significantly ($p = 0.02$), positively correlated with the log N:P (Fig. 1B). Foliar P concentration and N:P were significantly, negatively correlated ($p = 0.000$) with each other (Fig. 1C).

## Patterns of N and P and N:P in mineral soil and litter

Nitrogen concentrations and N:P varied markedly across mineral soil at 0–30 cm depth, ranging from $0.60 \pm 0.05$ mg g$^{-1}$ to $2.40 \pm 0.10$ mg g$^{-1}$ and $2.60 \pm 0.41$ to $6.81 \pm 0.51$, respectively (Table 4). However, P in mineral soil at 0–30 cm depth was significantly

**Table 2 DBH distribution of dominant tree species.** Big timber rate of domiant tree species was calculated in text.

| Mid-diameter (cm) | Ratio of tree species | | |
|---|---|---|---|
| | *Pinus tabuliformis* | *Pinus armandi* | *Quercus aliena* var. *acuteserrata* |
| 4 | 0.65 | 1.47 | 38.89 |
| 8 | 2.58 | 3.68 | 38.89 |
| 12 | 3.23 | 18.38 | 10.19 |
| 16 | 5.81 | 20.59 | 3.70 |
| 20 | 16.77 | 18.38 | 3.70 |
| 24 | 12.26 | 14.71 | 1.85 |
| 28 | 23.87 | 8.82 | 2.78 |
| 32 | 14.19 | 5.88 | |
| 36 | 10.32 | 5.15 | |
| 40 | 7.10 | 2.94 | |
| 44 | 2.58 | | |
| 48 | 0.65 | | |
| Sum | 100 | 100 | 100 |

**Table 3 Leaf stoichiometric traits of plants.**

| Tree species | TN (mg g$^{-1}$) | TP (mg g$^{-1}$) | N:P | IV (%) | Plants | | |
|---|---|---|---|---|---|---|---|
| | | | | | TN (mg g$^{-1}$) | TP (mg g$^{-1}$) | N:P ratio |
| *Pinus tabuliformis* | 11.84 ± 1.40[A] | 1.23 ± 0.22[a] | 9.62:1[a] | 35.2 | | | |
| *Pinus armandii* | 12.98 ± 1.43[B] | 1.20 ± 0.14[a] | 10.82:1[b] | 20.1 | 9.08 | 0.88 | 10.30:1 |
| *Quercus aliena* var. *acuteserrata* | 21.72 ± 3.19[C] | 2.00 ± 0.31[A] | 10.87:1[b] | 10.6 | | | |

**Notes.**

TN, TP, N:P and IV in the table represent leaf concentration of total nitrogen, phosphorus and N:P ratio both in dominant tree species and plants, and importance value of tree species, respectively.

[A]Extremely significant differences ($p = 0.000$) of leaf TN were between tree species.

[B]Extremely significant differences ($p = 0.000$) of leaf TP were between *Quercus aliena* var. *acuteserrata*-*Pinus tabuliformis* and Quercus aliena var. *acuteserrata*-*Pinus armandii*.

[C]Not significant difference ($p = 0.88$) of leaf TP was between *Pinus tabuliformis*-*Pinus armandii*.

[a]Significant differences of leaf N:P were between *Pinus tabuliformis*-*Pinus armandii* ($p = 0.048$) and *Pinus tabuliformis*-*Quercus aliena* var. *acuteserrata* ($p = 0.04$).

[b]Not significant difference ($p = 0.91$) of leaf N:P was between *Quercus aliena* var. *acuteserrata*-*Pinus armandii*.

different, ranging from $0.23 \pm 0.03$ mg g$^{-1}$ to $0.36 \pm 0.03$ mg g$^{-1}$ (Table 4). Bulk density of mineral soil increased with soil depth (Table 4). The indices, mean N:P and concentrations of N and P in litter were 2.49, 12.06 and 5.21 times of which in mineral soil respectively (Table 4).

Correlations between N and P concentrations in litter and mineral soil were complex and none was significant (Figs. 2A and 2B). At 0–10 cm depth, mineral soil N concentration increased with litter N concentration when litter N concentration was <17.9 mg g$^{-1}$, and vice versa (Fig. 2A). Soil P concentration increased with litter P concentration (Fig. 2B). At 11–20 cm depth, mineral soil N concentration decreased with litter N concentration when litter N concentration was <19.0 mg g$^{-1}$, and vice versa (Fig. 2A). Soil P concentration showed an exponential correlation with litter P concentration (Fig. 2B). At 21–30 cm depth, mineral soil N concentration increased with litter N concentration when litter N

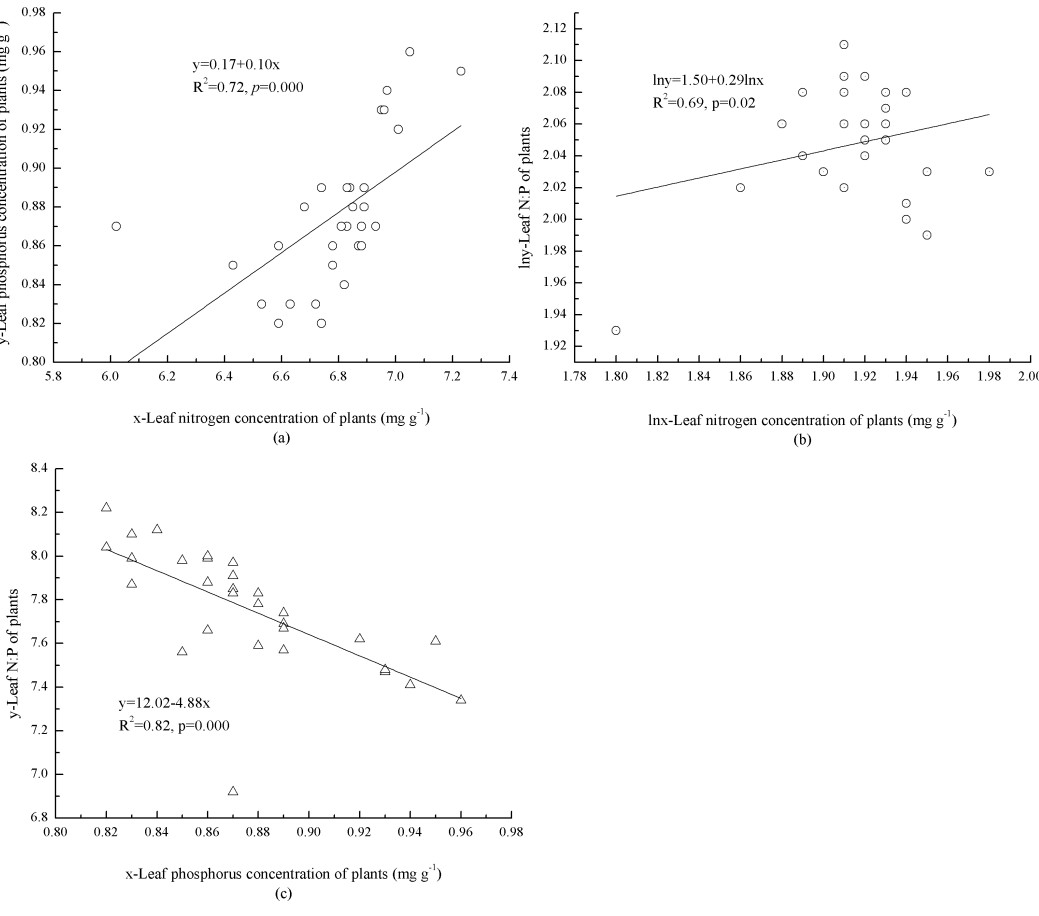

**Figure 1** **Interaction of foliar N, P concentrations and N:P ratio of plants.** (A) Leaf concentrations of N and P in plants. (B) Leaf concentration of N and N:P in plants. (C) Leaf concentration of P and N:P in plants.

**Table 4  Stoichiometric traits of mineral soil and litter.**

| Component | Depth (cm) | TN (mg g⁻¹) | TP (mg g⁻¹) | N:P ratio | Bulk density (g cm⁻³) | Biomass (t ha⁻¹) | Nutrients storage (t ha⁻¹) | |
|---|---|---|---|---|---|---|---|---|
| | | | | | | | TN | TP |
| Mineral soil | 0–10 | $2.40 \pm 0.10$ | $0.36 \pm 0.03$ | $(6.81:1) \pm 0.51$ | $1.04 \pm 0.02$ | | | |
| | 11–20 | $1.20 \pm 0.04$ | $0.27 \pm 0.03$ | $(4.45:1) \pm 0.39$ | $1.15 \pm 0.05$ | | $4.58 \pm 0.15$ | $0.95 \pm 0.09$ |
| | 21–30 | $0.60 \pm 0.05$ | $0.23 \pm 0.03$ | $(2.60:1) \pm 0.41$ | $1.22 \pm 0.06$ | | | |
| Mean | | 1.40 | 0.29 | 4.62 | | | | |
| Litter | | $16.89 \pm 3.59$ | $1.51 \pm 0.24$ | $(11.51:1) \pm 3.20$ | | $18.66 \pm 1.78$ | $0.31 \pm 0.07$ | $0.02 \pm 0.01$ |

Notes.
TN, TP and N:P in the table represent leaf concentration of total nitrogen, phosphorus and N:P ratio both in soil and litter, respectively.

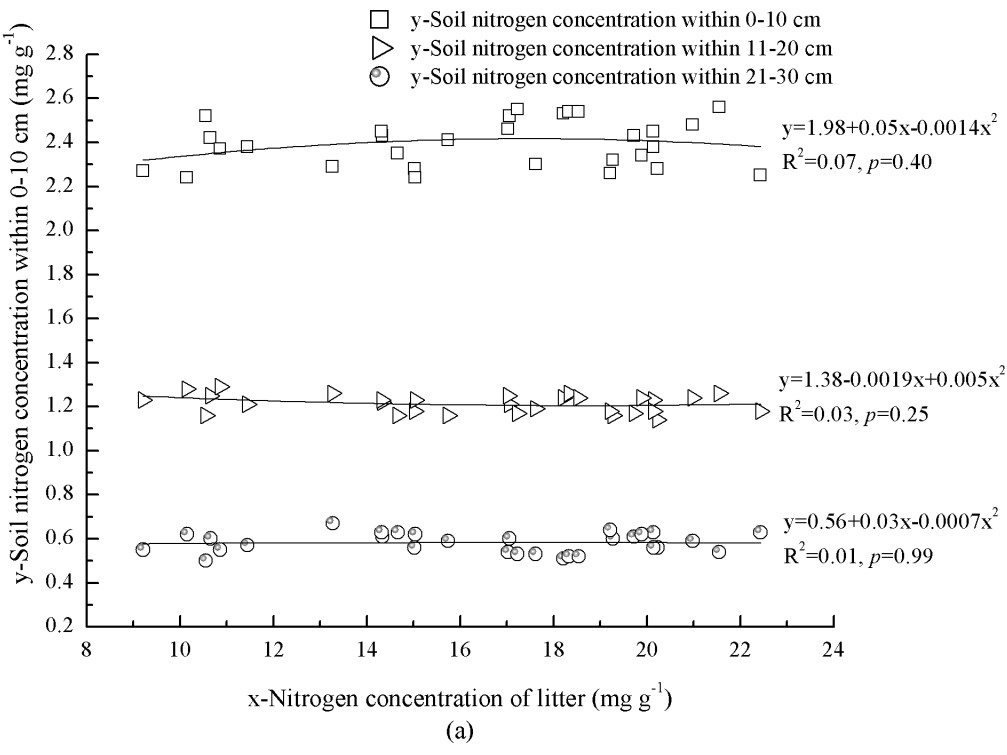

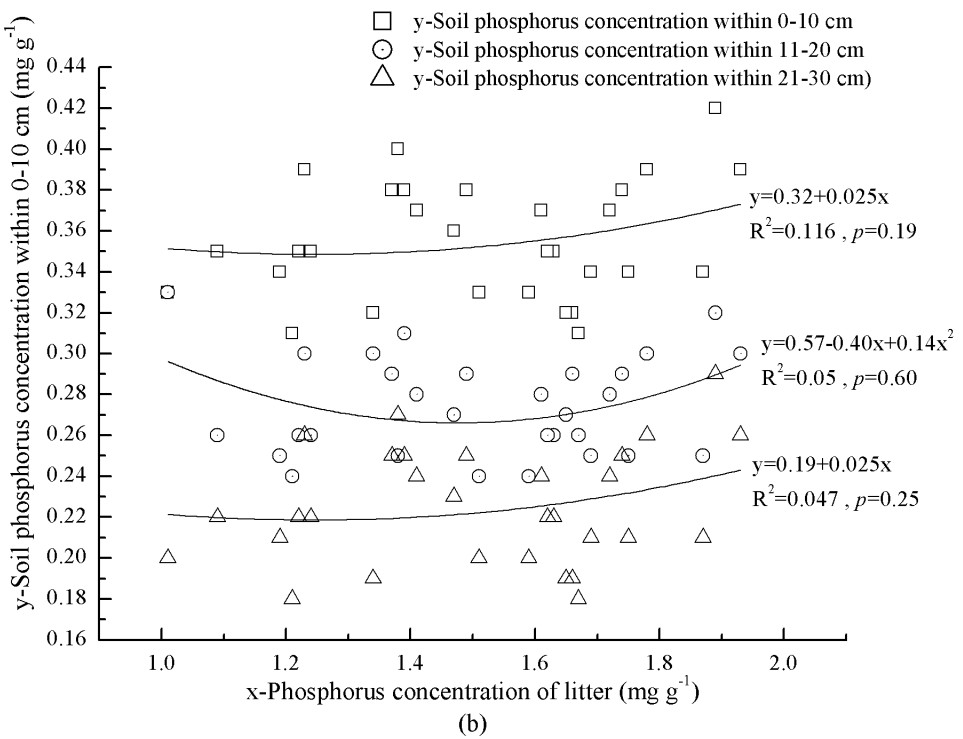

**Figure 2  Interaction of litter nutrients and mineral soil nutrients.** (A) Nitrogen concentrations of litter and miner soil within 0–10 cm. Nitrogen concentrations of litter and miner soil within 11–20 cm. Nitrogen concentrations of litter and miner soil within 21–30 cm. (B) Phosphorus concentrations of litter and miner soil within 0–10 cm. Phosphorus concentrations of litter and miner soil within 11–20 cm. Phosphorus concentrations of litter and miner soil within 21–30 cm.

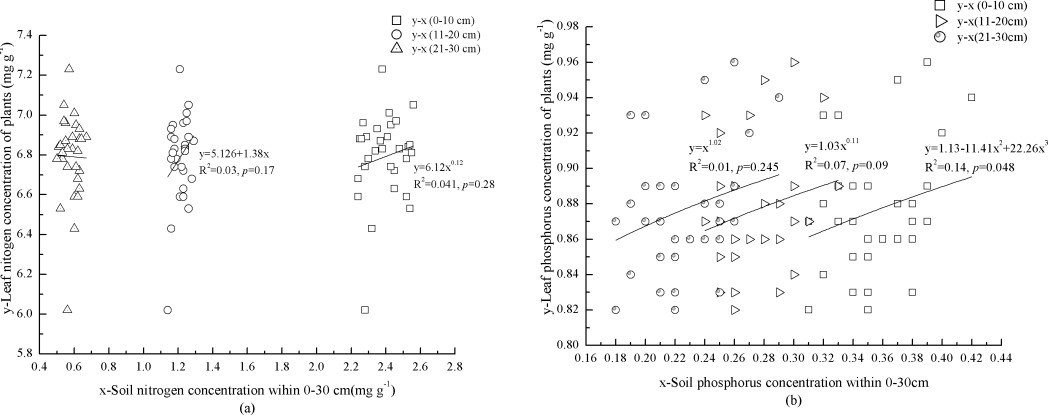

**Figure 3  Mineral soil nutrients and foliar nutrients of plants interaction.** Nitrogen concentrations of miner soil within 0–10 cm and leaf in plants. Nitrogen concentrations of miner soil within 11–20 cm and leaf in plants. Nitrogen concentrations of miner soil within 21–30 cm and leaf in plants. Phosphorus concentrations of miner soil within 0–10 cm and leaf in plants. Phosphorus concentrations of miner soil within 11–20 cm and leaf in plants. Phosphorus concentrations of miner soil within 21–30 cm and leaf in plants.

concentration was <17.9 mg g$^{-1}$, and vice versa (Fig. 2A). Phosphorus concentration of mineral soil increased linearly with litter P concentration (Fig. 2B).

## Correlation between foliar N and P concentration in the plants and mineral soil

Foliar N concentration in the plants increased with N concentration of mineral soil at 0–10 cm depth (Fig. 3A). In contrast, foliar P concentration in the plants showed a significant, cubic correlation with P concentration of mineral soil ($p = 0.048$) (Fig. 3B). Although foliar N and P concentrations in the plants were not significantly ($p = 0.09$) correlated with mineral soil at 11–20 cm depth, foliar N concentration in the plants increased with N concentration of mineral soil (Figs. 3A and 3B). There was no correlation between foliar N concentration in the plants and N concentration of mineral soil at 21–30 cm depth (Fig. 3A). Foliar P concentration in the plants increased with P concentration of mineral soil at 21–30 cm depth (Fig. 3B).

## DISCUSSION

### Foliar N and P stoichiometric traits in the plants in mixed pine-oak stand

The positive correlation between foliar N and P concentrations ($p = 0.000$, Fig. 1A) in the plants is consistent with stoichiometric stability criteria of a fixed ratio of nutrient absorption in the plant (*Koerselman & Meuleman, 1996*). Previously, mean foliar N and P and N:P were reported as 18.6, 1.21 mg g$^{-1}$, and 14.4, respectively for 753 species of terrestrial plants across China (*Han et al., 2005*) and 18.3, 1.42 mg g$^{-1}$, and 11.8, respectively for 1,251 world terrestrial plants (*Reich & Oleksyn, 2004*). This study indicated that foliar N:P of dominant tree species (*P. tabuliformis*, *P. armandii*, and *Quercus aliena*

var. *acuteserrata*.) and in the plants (10.30:1) were lower than both Chinese and global terrestrial plants. Similarly, foliar N concentration of *P. tabuliformis* ($11.84 \pm 1.40$ mg g$^{-1}$) and *P. armandii* ($12.98 \pm 1.43$ mg g$^{-1}$), and foliar P concentration of *P. armandii* ($1.20 \pm 0.14$ mg g$^{-1}$) were also lower. The overall element composition of plants in an ecosystem is determined by the mix of species and by the physiological status of the dominant plants (*Güsewell, 2004*). Therefore, potential explanations for the observed patterns are that foliar element concentrations and ratios were strongly determined by genetic and physiological controls, and that these crucial factors prevented plants from responding to the natural availability of nutrients (*Castle & Neff, 2009*). Furthermore, the nutrient status of terrestrial plants has a strong local and regional signal due to acquiring nutrients via weathering and microbial decomposition *in situ* (*Chadwick et al., 1999*). In the study area, low air temperature (<10 °C) (*Hou et al., 2017*) may have hindered rock weathering and microbial activity. However, the concentration of foliar N ($21.72 \pm 3.19$ mg g$^{-1}$) and P ($2.00 \pm 0.31$ mg g$^{-1}$) in *Q. aliena* var. *acuteserrata* was higher than that reported for Chinese and global mean levels. Foliar P concentration of *P. tabuliformis* ($1.23 \pm 0.22$ mg g$^{-1}$) was higher than the Chinese mean, but lower than the global mean. The growing period (from mid-May to late-September) in the study area was relatively short for plots on high elevation (more than 1,600 m) (*Hou et al., 2017*). The leaf life span of *P. tabuliformis* and *P. armandii* in the study area is 3 years. In the contrary, the leaf life span of *Q. aliena* var. *acuteserrata* is only 5 months. Therefore, shorter leaf life span and growing season may have caused higher N and P concentrations (*Castle & Neff, 2009*; *He et al., 2006*).

### Response of foliar nutrients in the plants to soil nutrients

The N and P concentrations in mineral soil were generally low and the similar result was also found by *Wu (2015)*. The N concentration in the top layer of mineral soil (0–10 cm) was above the standard of first class soil (>2.0 g kg$^{-1}$) (*National Soil Survey Office of China, 1998*); however, at 11–20 cm depth it only met the standard of third class soil (1–1.5 g kg$^{-1}$), and at 21–30 cm it met the standard of fifth class soil (0.5–0.75 g kg$^{-1}$) (*National Soil Survey Office of China, 1998*). The level of soil P concentration at 0–30 cm only reached the standard of third class soil (0.2–0.4 g kg$^{-1}$) (*National Soil Survey Office of China, 1998*). Litter decomposition and rock weathering are the main nutrient sources for mineral soil in natural forests. Plant element concentrations are largely determined by supplies of elements in soil and the chemical and physical characteristics of soil environments (*Castle & Neff, 2009*). However, our findings demonstrated that elements in litter did not significantly affect mineral soil. We found most of the correlations between foliar N and P concentrations in the plants and mineral soil at various depths were not significant, and soil nutrients did not explain more than 14% of foliar nutrients in the plants. Therefore, these results were not able to provide a better understanding and interpretation of the effects of litter and soil nutrients on mineral soil and foliar nutrients in the plants. Possible explanations for the observed patterns include high non-soluble chemical compounds in litter and low temperature (*Hou et al., 2017*), which slowed down litter decomposition. Furthermore, soil available N and P absorbed by plants have strong mobility and are easily

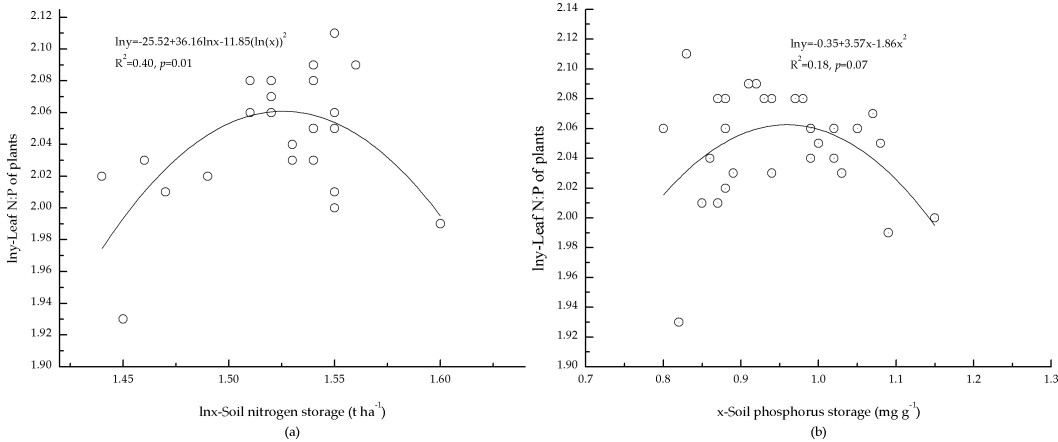

**Figure 4** **Relationships between storage of soil nutrients and foliar N:P ratio of plants.** (A) Relationship between nitrogen storage of mineral soil and N:P ratio of plants. (B) Relationship between phosphorus storage of mineral soil and N:P ratio of plants.

leached (*Thompson et al., 2010*). The thickness of mineral soil (at 30–50 cm depth), pore ratio from 32.0 to 62.28%, high precipitation (800–1,200 mm), and low temperature (*Hou et al., 2010*; *Hou et al., 2017*) in the study area may have accelerated leaching of N and P in decomposed litter and mineral soil, leaving less N and P to be assimilated into leaves of the plants. In addition, forest harvesting in 1950–1998 was also key cause to decrease N and P concentrations in mineral soil.

## Nutrient limitation of plants

It has been suggested that biomass N:P may be a better indicator of N or P deficiency than nutrient concentrations (*Güsewell, Koerselman & Verhoeven, 2003*). Previous studies have reported that plant growth is limited by concurrent N concentration <20 mg $g^{-1}$ and N:P < 14; while P limits plant growth at concentrations of <1 mg $g^{-1}$ and N:P >16. Therefore, co-limitation of N and P will occur when the concentrations and ratio of N and P meet these conditions (*Aerts & Chapin III, 1999*; *Koerselman & Meuleman, 1996*). The foliar N:P (10.3) and N concentration (9.08 mg $g^{-1}$) of the plants in the mixed pine-oak (Table 3) stand showed N limitation according to above mentioned criteria (*Aerts & Chapin III, 1999*; *Koerselman & Meuleman, 1996*).

The correlations between soil nutrient storage and foliar N:P of the plants indicated that foliar N:P of the plants was correlated with soil N storage (Fig. 4A). One possible reason may be that the growth of the plants was limited by N, and foliar N:P of the plants responses to soil N were more sensitive. However, N storage in mineral soil only explained 40% (Fig. 4A) of the variation in foliar N:P of the plants, and foliar N:P of the plants was not significantly correlated with soil P storage (Fig. 4B). This may have been a result of low P concentration in mineral soil, which also affected foliar N:P of the plants.

Our results showed that the total standing crop of elements in litter was estimated as 0.31 ± 0.07 t ha$^{-1}$ for N, and 0.02 ± 0.01 t ha$^{-1}$ for P; and 4.58 ± 0.15 t ha$^{-1}$ for N and 0.95 ± 0.09 t ha$^{-1}$ for P in mineral soil at 0–30 cm depth (Table 4). This suggested that trees

might assimilate N and P entirely from that stored in mineral soil and litter. According to the criteria (*Aerts & Chapin III, 1999*; *Koerselman & Meuleman, 1996*), 1.49 t ha$^{-1}$ pure N should be used to fertilize forest land to relieve N limitation on tree growth.

### Limitations of the analysis

We analyzed stoichiometric traits of leaves in the plants, mineral soil, and litter to explore the effects of N and P concentrations and N:P of litter and mineral soil on leaves in the plants. We also investigated the correlations between foliar N:P in the plants and N and P storage in mineral soil. The factors included in this study only explained 40% of the variation, which prevented us from fully understanding and interpreting total variation. Furthermore, besides criteria we used in the study, a more conservative estimate of N:P threshold is <10 for N limitation and >20 for P limitation (*Güsewell, 2004*; *Sardans, Rivasubach & Peñuelas, 2012*). Since there are multiple criteria for assessing nutrient limitation (*Aerts & Chapin III, 1999*; *Güsewell, 2004*; *Koerselman & Meuleman, 1996*; *Sardans, Rivasubach & Peñuelas, 2012*), we lack sufficient evidence to support an accurate estimation of fertilizer application rate. Finally, nutrient use efficiency and its influence on tree growth was not included in the estimation of pure N application rate, as it would have underestimated N. To verify these preliminary results, more study is required to detect the effects of fertilizer on stoichiometric traits of trees and mineral soil, and N and P interactions.

## CONCLUSIONS

Our results indicated that N and P concentrations were low in leaves in the plants and mineral soil, but high in litter. Concentrations of N and P in mineral soil were insufficient to explain the variation in foliar N:P in the plants. However, N storage in mineral soil at 0–30 cm depth was correlated with foliar N:P. The growth of the plants was limited by N; therefore, the stoichiometric approach we used in the present study reveals that approximately 1.49 t ha$^{-1}$ of N should be added if we want to achieve big timber cultivation. However, these rates should be further confirmed by a field fertilization experiments. We further recommend that appropriate fertilization rates should be included in the process of big timber cultivation. This result offers important insights into fertilizing forest land.

## ACKNOWLEDGEMENTS

We thank staff at Huoditang Forest Farm, Northwest A&F University for their assistance with sampling.

### Funding

This work was supported by National Key Research and Development Program of China (No. 2017YFD0600504-2) from Ministry of Science and Technology of the People's Republic of China. The funders had no role in study design, data collection and analysis, decision to publish, or preparation of the manuscript.

## Grant Disclosures

The following grant information was disclosed by the authors:
National Key Research and Development Program of China: 2017YFD0600504-2.
Ministry of Science and Technology of the People's Republic of China.

## Competing Interests

The authors declare there are no competing interests.

## Author Contributions

- Lin Hou conceived and designed the experiments, performed the experiments, prepared figures and/or tables, approved the final draft.
- Zhenjie Dong, Yuanyuan Yang and Donghong Zhang analyzed the data, contributed reagents/materials/analysis tools, approved the final draft.
- Shengli Zhang and Shuoxin Zhang authored or reviewed drafts of the paper, approved the final draft.

## Data Availability

The raw data are provided in File S1.

## Supplemental Information

Supplemental information for this article can be found online at http://dx.doi.org/10.7717/peerj.4628#supplemental-information.

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
