# Peer review of "Applying foliar stoichiometric traits of plants to determine fertilization for a mixed pine-oak stand in the Qinling Mountains, China"

_PeerJ, doi:10.7717/peerj.4628_

## Round 0.1 · original submission · Major Revisions

My suggested changes and reviewer comments are shown below and on your article 'Overview' screen.

Reviewer 1 ·

Basic reporting

The manuscript is devoted to N and P contents in forest stands with several tree species. N and P contents were determined in foliar, soil litter layer and in mineral soil samples. The authors used stoichiometry to asses N and P demand by forests and recommended a rate of fertilizer application. The most intriguing aspect for me was that despite of the fact the research sites are in the subtropics the stands appear to be N limited. Also interesting is the correlation between soil N and foliar N:P ratios, suggesting that soil N status affects tree P nutrition. I think that the paper contains enough interesting information and worth publishing after minor revision.

Experimental design

The experimental design is nice.

Validity of the findings

Interesting data

Additional comments

1) I would not recommend using the term “tree layer”, it sounds awkward. Better use “foliage” or vegetation, or plants instead.
2) The recommended N application rate seems too high. It should be compared to the rates applied in the region. Also important how you suggest to apply this amount of N: at once or distribute it across several years?
3) Line 34: not ”was added” but “should be added”
4) Line 185: Not relationship but correlations between
5) Line 195: Correlation, not relationship
6) Line 196: Replace the depth to the end of the sentence
7) Line 293: The recommendation of N fertilization must be softened. For example, the stoichiometric approach we used in the present study reveals that approximately 1.49 t/ha of N should be added if we want to achieve big timber cultivation. However, these rates should be further confirmed by a field fertilization experiments.

Reviewer 2 ·

Basic reporting

Several redundant figures such as figure 2 and 3. Author should consider another presentation methods.

Author should add the statistic analysis in Table 3 and also description in M&M section.

L224-229 How about N of P. tabuliformis and N and P of P. armandii ? Author should indicate the leaf life span of three species.

L261-267 There are arbitrary inferences in this paragraph. Need more evidences and supporting by other researches.

Experimental design

Need more detail information about species in M&M section.

L97-98 Author claim that the average stand age is 42 years. But the stand height only 9.2 m in average?

Validity of the findings

no comment

Additional comments

Several mistakes in unit should be corrected such as L25, L27,and L151.

The format of unit should be consistent such as mg.g-1 in Table 3 or mg g-1 in L187.

·

Basic reporting

My review with comments are attached

Experimental design

The experimental design was of high quality

Validity of the findings

See my review comments

Additional comments

The review contains my comments for the author. Additionally, Figures 2 and 3 need major revision. Although the research results contribute to current knowledge of N, P and N:P of forested ecosystems, the findings did not statistically support previously known behaviors of these elements in forested ecosystems of China. The authors should have concluded as such. All figure statistical p-values did not support the positive correlations the authors concurred.Trend lines or regression lines can point up or down but if they are not supported by statistical values, they do not mean much. The authors should not simply concur with previous conclusions elsewhere when they did an excellent work in a forested area whose climate and edaphic factors may have affected forest physiological ecology, and so the different results they found.

---

## Round 0.2 · accepted · Accept

Thank you for your revisions - the article is now Accepted

# Reviewer 1 ·

Basic reporting

The English is now sufficient. The text is well written and structured.

Experimental design

Excellent

Validity of the findings

Important and interesting findings

Additional comments

The authors have answered all my and other reviewers comments and properly addressed them in the text. Well done! I suggest to accept the manuscript for publishing.

Reviewer 2 ·

Basic reporting

Author has revised throughout the manuscript.
The line number is confused in WORD file. It's difficult to find the revised sentences.

Experimental design

I suggest that author should consider other statistical methods such as ANOVA and post hoc tests in Table 3.

Validity of the findings

no comment

Additional comments

How different is the upper/lowercase alphabet in Table 3?
According to the note description, it seems lowercase maybe enough. However, the footnote is too redundancy. The description in the last paragraph of M&M is enough.